# Scene Representation Networks: Continuous 3D-Structure-Aware Neural Scene Representations

**Vincent Sitzmann**     **Michael Zollhöfer**     **Gordon Wetzstein**

{sitzmann, zollhoefer}@cs.stanford.edu, gordon.wetzstein@stanford.edu
Stanford University

vsitzmann.github.io/srns/

## Abstract

Unsupervised learning with generative models has the potential of discovering rich representations of 3D scenes. While geometric deep learning has explored 3D-structure-aware representations of scene geometry, these models typically require explicit 3D supervision. Emerging neural scene representations can be trained only with posed 2D images, but existing methods ignore the three-dimensional structure of scenes. We propose Scene Representation Networks (SRNs), a continuous, 3D-structure-aware scene representation that encodes both geometry and appearance. SRNs represent scenes as continuous functions that map world coordinates to a feature representation of local scene properties. By formulating the image formation as a differentiable ray-marching algorithm, SRNs can be trained end-to-end from only 2D images and their camera poses, without access to depth or shape. This formulation naturally generalizes across scenes, learning powerful geometry and appearance priors in the process. We demonstrate the potential of SRNs by evaluating them for novel view synthesis, few-shot reconstruction, joint shape and appearance interpolation, and unsupervised discovery of a non-rigid face model.[1]

## 1   Introduction

A major driver behind recent work on generative models has been the promise of unsupervised discovery of powerful neural scene representations, enabling downstream tasks ranging from robotic manipulation and few-shot 3D reconstruction to navigation. A key aspect of solving these tasks is understanding the three-dimensional structure of an environment. However, prior work on neural scene representations either does not or only weakly enforces 3D structure [1–4]. Multi-view geometry and projection operations are performed by a black-box neural renderer, which is expected to learn these operations from data. As a result, such approaches fail to discover 3D structure under limited training data (see Sec. 4), lack guarantees on multi-view consistency of the rendered images, and learned representations are generally not interpretable. Furthermore, these approaches lack an intuitive interface to multi-view and projective geometry important in computer graphics, and cannot easily generalize to camera intrinsic matrices and transformations that were completely unseen at training time.

In geometric deep learning, many classic 3D scene representations, such as voxel grids [5–10], point clouds [11–14], or meshes [15] have been integrated with end-to-end deep learning models and have led to significant progress in 3D scene understanding. However, these scene representations are discrete, limiting achievable spatial resolution, only sparsely sampling the underlying smooth surfaces of a scene, and often require explicit 3D supervision.

We introduce *Scene Representation Networks* (SRNs), a continuous neural scene representation, along with a differentiable rendering algorithm, that model both 3D scene geometry and appearance, enforce 3D structure in a multi-view consistent manner, and naturally allow generalization of shape and appearance priors across scenes. The key idea of SRNs is to represent a scene implicitly as a continuous, differentiable function that maps a 3D world coordinate to a feature-based representation of the scene properties at that coordinate. This allows SRNs to naturally interface with established techniques of multi-view and projective geometry while operating at high spatial resolution in a memory-efficient manner. SRNs can be trained end-to-end, supervised only by a set of posed 2D images of a scene. SRNs generate high-quality images *without any 2D convolutions*, exclusively operating on individual pixels, which enables image generation at arbitrary resolutions. They generalize naturally to camera transformations and intrinsic parameters that were completely unseen at training time. For instance, SRNs that have only ever seen objects from a constant distance are capable of rendering close-ups of said objects flawlessly. We evaluate SRNs on a variety of challenging 3D computer vision problems, including novel view synthesis, few-shot scene reconstruction, joint shape and appearance interpolation, and unsupervised discovery of a non-rigid face model.

To summarize, our approach makes the following key contributions:

- A continuous, 3D-structure-aware neural scene representation and renderer, SRNs, that efficiently encapsulate both scene geometry and appearance.
- End-to-end training of SRNs without explicit supervision in 3D space, purely from a set of posed 2D images.
- We demonstrate novel view synthesis, shape and appearance interpolation, and few-shot reconstruction, as well as unsupervised discovery of a non-rigid face model, and significantly outperform baselines from recent literature.

**Scope**   The current formulation of SRNs does not model view- and lighting-dependent effects or translucency, reconstructs shape and appearance in an entangled manner, and is non-probabilistic. Please see Sec. 5 for a discussion of future work in these directions.

## 2   Related Work

Our approach lies at the intersection of multiple fields. In the following, we review related work.

**Geometric Deep Learning.**   Geometric deep learning has explored various representations to reason about scene geometry. Discretization-based techniques use voxel grids [7, 16–22], octree hierarchies [23–25], point clouds [11, 26, 27], multiplane images [28], patches [29], or meshes [15, 21, 30, 31]. Methods based on function spaces continuously represent space as the decision boundary of a learned binary classifier [32] or a continuous signed distance field [33–35]. While these techniques are successful at modeling geometry, they often require 3D supervision, and it is unclear how to efficiently infer and represent appearance. Our proposed method encapsulates both scene geometry and appearance, and can be trained end-to-end via learned differentiable rendering, supervised only with posed 2D images.

**Neural Scene Representations.**   Latent codes of autoencoders may be interpreted as a feature representation of the encoded scene. Novel views may be rendered by concatenating target pose and latent code [1] or performing view transformations directly in the latent space [4]. Generative Query Networks [2, 3] introduce a probabilistic reasoning framework that models uncertainty due to incomplete observations, but both the scene representation and the renderer are oblivious to the scene's 3D structure. Some prior work infers voxel grid representations of 3D scenes from images [6, 8, 9] or uses them for 3D-structure-aware generative models [10, 36]. Graph neural networks may similarly capture 3D structure [37]. Compositional structure may be modeled by representing scenes as programs [38]. We demonstrate that models with scene representations that ignore 3D structure fail to perform viewpoint transformations in a regime of limited (but significant) data, such as the Shapenet v2 dataset [39]. Instead of a discrete representation, which limits achievable spatial resolution and does not smoothly parameterize scene surfaces, we propose a continuous scene representation.

**Neural Image Synthesis.**   Deep models for 2D image and video synthesis have recently shown promising results in generating photorealistic images. Some of these approaches are based on

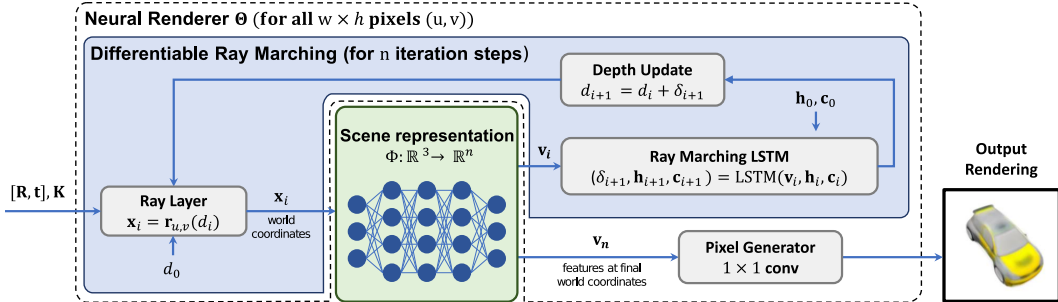

Figure 1: Overview: at the heart of SRNs lies a continuous, 3D-aware neural scene representation, $\Phi$, which represents a scene as a function that maps $(x, y, z)$ world coordinates to a feature representation of the scene at those coordinates (see Sec. 3.1). A neural renderer $\Theta$, consisting of a learned ray marcher and a pixel generator, can render the scene from arbitrary novel view points (see Sec. 3.2).

(variational) auto-encoders [40, 41], generative flows [42, 43], or autoregressive per-pixel models [44, 45]. In particular, generative adversarial networks [46–50] and their conditional variants [51–53] have recently achieved photo-realistic single-image generation. Compositional Pattern Producing Networks [54, 55] learn functions that map 2D image coordinates to color. Some approaches build on explicit spatial or perspective transformations in the networks [56–58, 14]. Recently, following the spirit of "vision as inverse graphics" [59, 60], deep neural networks have been applied to the task of inverting graphics engines [61–65]. However, these 2D generative models only learn to parameterize the manifold of 2D natural images, and struggle to generate images that are multi-view consistent, since the underlying 3D scene structure cannot be exploited.

## 3 Formulation

Given a training set $\mathcal{C} = \{(\mathcal{I}_i, \mathbf{E}_i, \mathbf{K}_i)\}_{i=1}^N$ of $N$ tuples of images $\mathcal{I}_i \in \mathbb{R}^{H \times W \times 3}$ along with their respective extrinsic $\mathbf{E}_i = [\mathbf{R}|\mathbf{t}] \in \mathbb{R}^{3 \times 4}$ and intrinsic $\mathbf{K}_i \in \mathbb{R}^{3 \times 3}$ camera matrices [66], our goal is to distill this dataset of observations into a neural scene representation $\Phi$ that strictly enforces 3D structure and allows to generalize shape and appearance priors across scenes. In addition, we are interested in a rendering function $\Theta$ that allows us to render the scene represented by $\Phi$ from arbitrary viewpoints. In the following, we first formalize $\Phi$ and $\Theta$ and then discuss a framework for optimizing $\Phi$, $\Theta$ for a single scene given only posed 2D images. Note that this approach does *not* require information about scene geometry. Additionally, we show how to learn a family of scene representations for an entire class of scenes, discovering powerful shape and appearance priors.

### 3.1 Representing Scenes as Functions

Our key idea is to represent a scene as a function $\Phi$ that maps a spatial location $\mathbf{x}$ to a feature representation $\mathbf{v}$ of learned scene properties at that spatial location:

$$\Phi : \mathbb{R}^3 \to \mathbb{R}^n, \quad \mathbf{x} \mapsto \Phi(\mathbf{x}) = \mathbf{v}. \qquad (1)$$

The feature vector $\mathbf{v}$ may encode visual information such as surface color or reflectance, but it may also encode higher-order information, such as the signed distance of $\mathbf{x}$ to the closest scene surface. This continuous formulation can be interpreted as a generalization of discrete neural scene representations. Voxel grids, for instance, discretize $\mathbb{R}^3$ and store features in the resulting 3D grid [5–10]. Point clouds [12–14] may contain points at any position in $\mathbb{R}^3$, but only sparsely sample surface properties of a scene. In contrast, $\Phi$ densely models scene properties and can in theory model arbitrary spatial resolutions, as it is continuous over $\mathbb{R}^3$ and can be sampled with arbitrary resolution. In practice, we represent $\Phi$ as a multi-layer perceptron (MLP), and spatial resolution is thus limited by the capacity of the MLP.

In contrast to recent work on representing scenes as unstructured or weakly structured feature embeddings [1, 4, 2], $\Phi$ is explicitly aware of the 3D structure of scenes, as the input to $\Phi$ are world coordinates $(x, y, z) \in \mathbb{R}^3$. This allows interacting with $\Phi$ via the toolbox of multi-view and perspective geometry that the physical world obeys, only using learning to approximate the unknown properties of the scene itself. In Sec. 4, we show that this formulation leads to multi-view consistent novel view synthesis, data-efficient training, and a significant gain in model interpretability.

## 3.2 Neural Rendering

Given a scene representation $\Phi$, we introduce a neural rendering algorithm $\Theta$, that maps a scene representation $\Phi$ as well as the intrinsic $\mathbf{K}$ and extrinsic $\mathbf{E}$ camera parameters to an image $\mathcal{I}$:

$$\Theta : \mathcal{X} \times \mathbb{R}^{3\times 4} \times \mathbb{R}^{3\times 3} \to \mathbb{R}^{H\times W\times 3}, \quad (\Phi, \mathbf{E}, \mathbf{K}) \mapsto \Theta(\Phi, \mathbf{E}, \mathbf{K}) = \mathcal{I}, \tag{2}$$

where $\mathcal{X}$ is the space of all functions $\Phi$.

The key complication in rendering a scene represented by $\Phi$ is that geometry is represented implicitly. The surface of a wooden table top, for instance, is defined by the subspace of $\mathbb{R}^3$ where $\Phi$ undergoes a change from a feature vector representing free space to one representing wood.

To render a single pixel in the image observed by a virtual camera, we thus have to solve two sub-problems: (i) finding the world coordinates of the intersections of the respective camera rays with scene geometry, and (ii) mapping the feature vector $\mathbf{v}$ at that spatial coordinate to a color. We will first propose a neural ray marching algorithm with learned, adaptive step size to find ray intersections with scene geometry, and subsequently discuss the architecture of the pixel generator network that learns the feature-to-color mapping.

### 3.2.1 Differentiable Ray Marching Algorithm

---
**Algorithm 1** Differentiable Ray-Marching
---
1: **function** FINDINTERSECTION($\Phi, \mathbf{K}, \mathbf{E}, (u, v)$)
2:     $d_0 \leftarrow 0.05$                                                           ▷ Near plane
3:     $(\mathbf{h}_0, \mathbf{c}_0) \leftarrow (\mathbf{0}, \mathbf{0})$                 ▷ Initial state of LSTM
4:     **for** $i \leftarrow 0$ to $max\_iter$ **do**
5:         $\mathbf{x}_i \leftarrow \mathbf{r}_{u,v}(d_i)$                               ▷ Calculate world coordinates
6:         $\mathbf{v}_i \leftarrow \Phi(\mathbf{x}_i)$                                 ▷ Extract feature vector
7:         $(\delta, \mathbf{h}_{i+1}, \mathbf{c}_{i+1}) \leftarrow LSTM(\mathbf{v}, \mathbf{h}_i, \mathbf{c}_i)$   ▷ Predict steplength using ray marching LSTM
8:         $d_{i+1} \leftarrow d_i + \delta$                                           ▷ Update d
9:     **return** $\mathbf{r}_{u,v}(d_{max\_iter})$
---

Intersection testing intuitively amounts to solving an optimization problem, where the point along each camera ray is sought that minimizes the distance to the surface of the scene. To model this problem, we parameterize the points along each ray, identified with the coordinates $(u, v)$ of the respective pixel, with their distance $d$ to the camera ($d > 0$ represents points in front of the camera):

$$\mathbf{r}_{u,v}(d) = \mathbf{R}^T (\mathbf{K}^{-1} \begin{pmatrix} u \\ v \\ d \end{pmatrix} - \mathbf{t}), \quad d > 0, \tag{3}$$

with world coordinates $\mathbf{r}_{u,v}(d)$ of a point along the ray with distance $d$ to the camera, camera intrinsics $\mathbf{K}$, and camera rotation matrix $\mathbf{R}$ and translation vector $\mathbf{t}$. For each ray, we aim to solve

$$\begin{aligned} \arg\min \quad & d \\ \text{s.t.} \quad & \mathbf{r}_{u,v}(d) \in \Omega, \quad d > 0 \end{aligned} \tag{4}$$

where we define the set of all points that lie on the surface of the scene as $\Omega$.

Here, we take inspiration from the classic sphere tracing algorithm [67]. Sphere tracing belongs to the class of ray marching algorithms, which solve Eq. 4 by starting at a distance $d_{init}$ close to the camera and stepping along the ray until scene geometry is intersected. Sphere tracing is defined by a special choice of the step length: each step has a length equal to the signed distance to the closest surface point of the scene. Since this distance is only 0 on the surface of the scene, the algorithm takes non-zero steps until it has arrived at the surface, at which point no further steps are taken. Extensions of this algorithm propose heuristics to modifying the step length to speed up convergence [68]. We instead propose to *learn* the length of each step.

Specifically, we introduce a *ray marching long short-term memory (RM-LSTM)* [69], that maps the feature vector $\Phi(\mathbf{x}_i) = \mathbf{v}_i$ at the current estimate of the ray intersection $\mathbf{x}_i$ to the length of the next ray marching step. The algorithm is formalized in Alg. 1.

Given our current estimate $d_i$, we compute world coordinates $\mathbf{x}_i = \mathbf{r}_{u,v}(d_i)$ via Eq. 3. We then compute $\Phi(\mathbf{x}_i)$ to obtain a feature vector $\mathbf{v}_i$, which we expect to encode information about nearby scene surfaces. We then compute the step length $\delta$ via the RM-LSTM as $(\delta, \mathbf{h}_{i+1}, \mathbf{c}_{i+1}) = LSTM(\mathbf{v}_i, \mathbf{h}_i, \mathbf{c}_i)$, where $\mathbf{h}$ and $\mathbf{c}$ are the output and cell states, and increment $d_i$ accordingly. We iterate this process for a constant number of steps. This is critical, because a dynamic termination criterion would have no guarantee for convergence in the beginning of the training, where both $\Phi$ and the ray marching LSTM are initialized at random. The final step yields our estimate of the world coordinates of the intersection of the ray with scene geometry. The $z$-coordinates of running and final estimates of intersections in camera coordinates yield depth maps, which we denote as $\mathbf{d}_i$, which visualize every step of the ray marcher. This makes the ray marcher interpretable, as failures in geometry estimation show as inconsistencies in the depth map. Note that depth maps are differentiable with respect to all model parameters, but are not required for training $\Phi$. Please see the supplement for a contextualization of the proposed rendering approach with classical rendering algorithms.

### 3.2.2 Pixel Generator Architecture

The pixel generator takes as input the 2D feature map sampled from $\Phi$ at world coordinates of ray-surface intersections and maps it to an estimate of the observed image. As a generator architecture, we choose a per-pixel MLP that maps a single feature vector $\mathbf{v}$ to a single RGB vector. This is equivalent to a convolutional neural network (CNN) with only $1 \times 1$ convolutions. Formulating the generator without 2D convolutions has several benefits. First, the generator will always map the same $(x, y, z)$ coordinate to the same color value. Assuming that the ray-marching algorithm finds the correct intersection, the rendering is thus trivially multi-view consistent. This is in contrast to 2D convolutions, where the value of a single pixel depends on a neighborhood of features in the input feature map. When transforming the camera in 3D, e.g. by moving it closer to a surface, the 2D neighborhood of a feature may change. As a result, 2D convolutions come with no guarantee on multi-view consistency. With our per-pixel formulation, the rendering function $\Theta$ operates independently on all pixels, allowing images to be generated with arbitrary resolutions and poses. On the flip side, we cannot exploit recent architectural progress in CNNs, and a per-pixel formulation requires the ray marching, the SRNs and the pixel generator to operate on the same (potentially high) resolution, requiring a significant memory budget. Please see the supplement for a discussion of this trade-off.

### 3.3 Generalizing Across Scenes

We now generalize SRNs from learning to represent a single scene to learning shape and appearance priors over several instances of a single class. Formally, we assume that we are given a set of $M$ instance datasets $\mathcal{D} = \{\mathcal{C}_j\}_{j=1}^M$, where each $\mathcal{C}_j$ consists of tuples $\{(\mathcal{I}_i, \mathbf{E}_i, \mathbf{K}_i)\}_{i=1}^N$ as discussed in Sec. 3.1.

We reason about the set of functions $\{\Phi_j\}_{j=1}^M$ that represent instances of objects belonging to the same class. By parameterizing a specific $\Phi_j$ as an MLP, we can represent it with its vector of parameters $\phi_j \in \mathbb{R}^l$. We assume scenes of the same class have common shape and appearance properties that can be fully characterized by a set of latent variables $\mathbf{z} \in \mathbb{R}^k$, $k < l$. Equivalently, this assumes that all parameters $\phi_j$ live in a $k$-dimensional subspace of $\mathbb{R}^l$. Finally, we define a mapping

$$\Psi : \mathbb{R}^k \to \mathbb{R}^l, \quad \mathbf{z}_j \mapsto \Psi(\mathbf{z}_j) = \phi_j \tag{5}$$

that maps a latent vector $\mathbf{z}_j$ to the parameters $\phi_j$ of the corresponding $\Phi_j$. We propose to parameterize $\Psi$ as an MLP, with parameters $\psi$. This architecture was previously introduced as a Hypernetwork [70], a neural network that regresses the parameters of another neural network. We share the parameters of the rendering function $\Theta$ across scenes. We note that assuming a low-dimensional embedding manifold has so far mainly been empirically demonstrated for classes of single objects. Here, we similarly only demonstrate generalization over classes of single objects.

**Finding latent codes $\mathbf{z}_j$.** To find the latent code vectors $\mathbf{z}_j$, we follow an auto-decoder framework [33]. For this purpose, each object instance $\mathcal{C}_j$ is represented by its own latent code $\mathbf{z}_j$. The $\mathbf{z}_j$ are free variables and are optimized jointly with the parameters of the hypernetwork $\Psi$ and the neural renderer $\Theta$. We assume that the prior distribution over the $\mathbf{z}_j$ is a zero-mean multivariate Gaussian with a diagonal covariance matrix. Please refer to [33] for additional details.

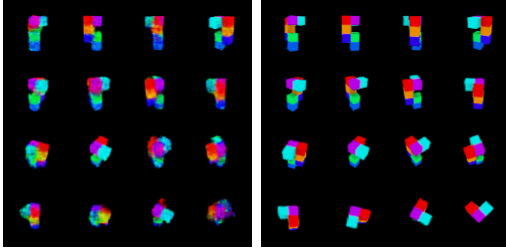
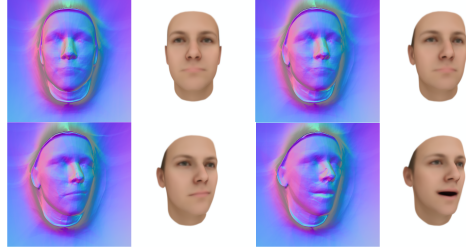

Figure 2: Shepard-Metzler object from 1k-object training set, 15 observations each. SRNs (right) outperform dGQN (left) on this small dataset.

Figure 3: Non-rigid animation of a face. Note that mouth movement is directly reflected in the normal maps.

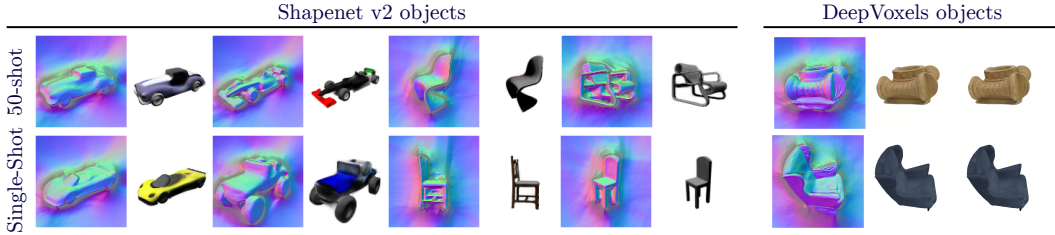

Figure 4: Normal maps for a selection of objects. We note that geometry is learned fully unsupervised and arises purely out of the perspective and multi-view geometry constraints on the image formation.

## 3.4 Joint Optimization

To summarize, given a dataset $\mathcal{D} = \{\mathcal{C}_j\}_{j=1}^M$ of instance datasets $\mathcal{C} = \{(\mathcal{I}_i, \mathbf{E}_i, \mathbf{K}_i)\}_{i=1}^N$, we aim to find the parameters $\psi$ of $\Psi$ that maps latent vectors $\mathbf{z}_j$ to the parameters of the respective scene representation $\phi_j$, the parameters $\theta$ of the neural rendering function $\Theta$, as well as the latent codes $\mathbf{z}_j$ themselves. We formulate this as an optimization problem with the following objective:

$$\underset{\{\theta,\psi,\{\mathbf{z}_j\}_{j=1}^M\}}{\arg\min} \sum_{j=1}^M \sum_{i=1}^N \underbrace{\|\Theta_\theta(\Phi_{\Psi(\mathbf{z_j})}, \mathbf{E}_i^j, \mathbf{K}_i^j) - \mathcal{I}_i^j\|_2^2}_{\mathcal{L}_{\text{img}}} + \underbrace{\lambda_{dep}\|\min(\mathbf{d}_{i,final}^j, \mathbf{0})\|_2^2}_{\mathcal{L}_{\text{depth}}} + \underbrace{\lambda_{lat}\|\mathbf{z}_j\|_2^2}_{\mathcal{L}_{\text{latent}}}. \quad (6)$$

Where $\mathcal{L}_{\text{img}}$ is an $\ell_2$-loss enforcing closeness of the rendered image to ground-truth, $\mathcal{L}_{\text{depth}}$ is a regularization term that accounts for the positivity constraint in Eq. 4, and $\mathcal{L}_{\text{latent}}$ enforces a Gaussian prior on the $\mathbf{z}_j$. In the case of a single scene, this objective simplifies to solving for the parameters $\phi$ of the MLP parameterization of $\Phi$ instead of the parameters $\psi$ and latent codes $\mathbf{z}_j$. We solve Eq. 6 with stochastic gradient descent. Note that the whole pipeline can be trained end-to-end, without requiring any (pre-)training of individual parts. In Sec. 4, we demonstrate that SRNs discover both geometry and appearance, initialized at random, without requiring prior knowledge of either scene geometry or scene scale, enabling multi-view consistent novel view synthesis.

**Few-shot reconstruction.** After finding model parameters by solving Eq. 6, we may use the trained model for few-shot reconstruction of a new object instance, represented by a dataset $\mathcal{C} = \{(\mathcal{I}_i, \mathbf{E}_i, \mathbf{K}_i)\}_{i=1}^N$. We fix $\theta$ as well as $\psi$, and estimate a new latent code $\hat{\mathbf{z}}$ by minimizing

$$\hat{\mathbf{z}} = \underset{\mathbf{z}}{\arg\min} \sum_{i=1}^N \|\Theta_\theta(\Phi_{\Psi(\mathbf{z})}, \mathbf{E}_i, \mathbf{K}_i) - \mathcal{I}_i\|_2^2 + \lambda_{dep}\|\min(\mathbf{d}_{i,final}, \mathbf{0})\|_2^2 + \lambda_{lat}\|\mathbf{z}\|_2^2 \quad (7)$$

## 4 Experiments

We train SRNs on several object classes and evaluate them for novel view synthesis and few-shot reconstruction. We further demonstrate the discovery of a non-rigid face model. Please see the supplement for a comparison on single-scene novel view synthesis performance with DeepVoxels [6].

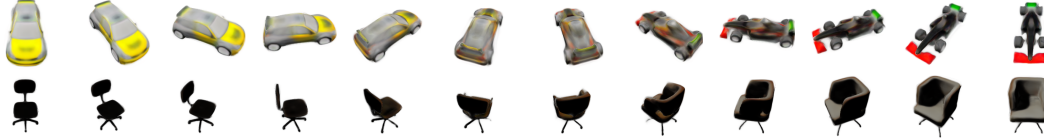

Figure 5: Interpolating latent code vectors of cars and chairs in the Shapenet dataset while rotating the camera around the model. Features smoothly transition from one model to another.

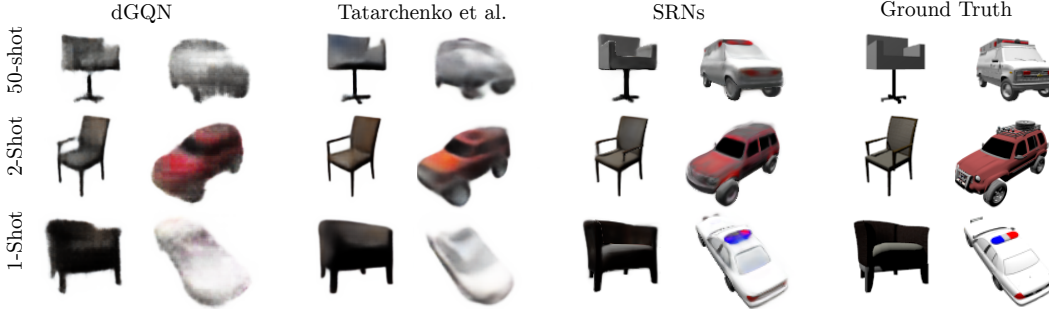

Figure 6: Qualitative comparison with Tatarchenko et al. [1] and the deterministic variant of the GQN [2], for novel view synthesis on the Shapenet v2 "cars" and "chairs" classes. We compare novel views for objects reconstructed from 50 observations in the training set (top row), two observations and a single observation (second and third row) from a test set. SRNs consistently outperforms these baselines with multi-view consistent novel views, while also reconstructing geometry. Please see the supplemental video for more comparisons, smooth camera trajectories, and reconstructed geometry.

**Implementation Details.** Hyperparameters, computational complexity, and full network architectures for SRNs and all baselines are in the supplement. Training of the presented models takes on the order of 6 days. A single forward pass takes around 120 ms and 3 GB of GPU memory per batch item. Code and datasets are available.

**Shepard-Metzler objects.** We evaluate our approach on 7-element Shepard-Metzler objects in a limited-data setting. We render 15 observations of 1k objects at a resolution of $64 \times 64$. We train both SRNs and a deterministic variant of the Generative Query Network [2] (dGQN, please see supplement for an extended discussion). Note that the dGQN is solving a harder problem, as it is inferring the scene representation in each forward pass, while our formulation requires solving an optimization problem to find latent codes for unseen objects. We benchmark novel view reconstruction accuracy on (1) the training set and (2) few-shot reconstruction of 100 objects from a held-out test set. On the training objects, SRNs achieve almost pixel-perfect results with a PSNR of 30.41 dB. The dGQN fails to learn object shape and multi-view geometry on this limited dataset, achieving 20.85 dB. See Fig. 2 for a qualitative comparison. In a two-shot setting (see Fig. 7 for reference views), we succeed in reconstructing any part of the object that has been observed, achieving 24.36 dB, while the dGQN achieves 18.56 dB. In a one-shot setting, SRNs reconstruct an object consistent with the observed view. As expected, due to the current non-probabilistic implementation, both the dGQN and SRNs reconstruct an object resembling the mean of the hundreds of feasible objects that may have generated the observation, achieving 17.51 dB and 18.11 dB respectively.

**Shapenet v2.** We consider the "chair" and "car" classes of Shapenet v.2 [39] with 4.5k and 2.5k model instances respectively. We disable transparencies and specularities, and train on 50 observations of each instance at a resolution of $128 \times 128$ pixels. Camera poses are randomly generated on a sphere with the object at the origin. We evaluate performance on (1) novel-view synthesis of objects in the training set and (2) novel-view synthesis on objects in the held-out, official Shapenet v2 test sets, reconstructed from one or two observations, as discussed in Sec. 3.4.

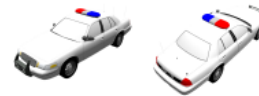

Figure 7: Single- (left) and two-shot (both) reference views.

Fig. 7 shows the sampled poses for the few-shot case. In all settings, we assemble ground-truth novel views by sampling 250 views in an Archimedean spiral around each object instance. We compare

Table 1: PSNR (in dB) and SSIM of images reconstructed with our method, the deterministic variant of the GQN [2] (dGQN), the model proposed by Tatarchenko et al. [1] (TCO), and the method proposed by Worrall et al. [4] (WRL). We compare novel-view synthesis performance on objects in the training set (containing 50 images of each object), as well as reconstruction from 1 or 2 images on the held-out test set.

|  | 50 images (training set) | | 2 images | | Single image | |
|---|---|---|---|---|---|---|
|  | Chairs | Cars | Chairs | Cars | Chairs | Cars |
| TCO [1] | 24.31 / 0.92 | 20.38 / 0.83 | 21.33 / 0.88 | 18.41 / 0.80 | 21.27 / 0.88 | 18.15 / 0.79 |
| WRL [4] | 24.57 / 0.93 | 19.16 / 0.82 | 22.28 / 0.90 | 17.20 / 0.78 | 22.11 / 0.90 | 16.89 / 0.77 |
| dGQN [2] | 22.72 / 0.90 | 19.61 / 0.81 | 22.36 / 0.89 | 18.79 / 0.79 | 21.59 / 0.87 | 18.19 / 0.78 |
| SRNs | **26.23 / 0.95** | **26.32 / 0.94** | **24.48 / 0.92** | **22.94 / 0.88** | **22.89 / 0.91** | **20.72 / 0.85** |

SRNs to three baselines from recent literature. Table 1 and Fig. 6 report quantitative and qualitative results respectively. In all settings, we outperform all baselines by a wide margin. On the training set, we achieve very high visual fidelity. Generally, views are perfectly multi-view consistent, the only exception being objects with distinct, usually fine geometric detail, such as the windscreen of convertibles. None of the baselines succeed in generating multi-view consistent views. Several views per object are usually entirely degenerate. In the two-shot case, where most of the object has been seen, SRNs still reconstruct both object appearance and geometry robustly. In the single-shot case, SRNs complete unseen parts of the object in a plausible manner, demonstrating that the learned priors have truthfully captured the underlying distributions.

**Supervising parameters for non-rigid deformation.** If latent parameters of the scene are known, we can condition on these parameters instead of jointly solving for latent variables $\mathbf{z}_j$. We generate 50 renderings each from 1000 faces sampled at random from the Basel face model [71]. Camera poses are sampled from a hemisphere in front of the face. Each face is fully defined by a 224-dimensional parameter vector, where the first 160 parameterize identity, and the last 64 dimensions control facial expression. We use a constant ambient illumination to render all faces. Conditioned on this disentangled latent space, SRNs succeed in reconstructing face geometry and appearance. After training, we animate facial expression by varying the 64 expression parameters while keeping the identity fixed, even though this specific combination of identity and expression has not been observed before. Fig. 3 shows qualitative results of this non-rigid deformation. Expressions smoothly transition from one to the other, and the reconstructed normal maps, which are directly computed from the depth maps (not shown), demonstrate that the model has learned the underlying geometry.

**Geometry reconstruction.** SRNs reconstruct geometry in a fully unsupervised manner, purely out of necessity to explain observations in 3D. Fig. 4 visualizes geometry for 50-shot, single-shot, and single-scene reconstructions.

**Latent space interpolation.** Our learned latent space allows meaningful interpolation of object instances. Fig. 5 shows latent space interpolation.

**Pose extrapolation.** Due to the explicit 3D-aware and per-pixel formulation, SRNs naturally generalize to 3D transformations that have never been seen during training, such as camera close-ups or camera roll, even when trained only on up-right camera poses distributed on a sphere around the objects. Please see the supplemental video for examples of pose extrapolation.

**Failure cases.** The ray marcher may "get stuck" in holes of surfaces or on rays that closely pass by occluders, such as commonly occur in chairs. SRNs generates a continuous surface in these cases, or will sometimes step through the surface. If objects are far away from the training distribution, SRNs may fail to reconstruct geometry and instead only match texture. In both cases, the reconstructed geometry allows us to analyze the failure, which is impossible with black-box alternatives. See Fig. 8 and the supplemental video.

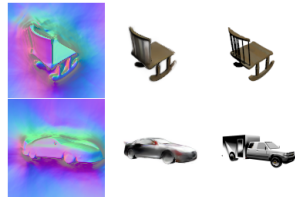

Figure 8: Failure cases.

**Towards representing room-scale scenes.** We demonstrate reconstruction of a room-scale scene with SRNs. We train a single SRN on 500 observations of a minecraft room. The room contains multiple objects as well as four columns, such that parts of the scene are occluded in most observations. After training, the SRN enables novel view synthesis of the room. Though generated images are blurry, they are largely multi-view consistent, with artifacts due to ray marching failures only at object boundaries and thin structures. The SRN succeeds in inferring geometry and appearance of the room, reconstructing occluding columns and objects correctly, failing only on low-texture areas (where geometry is only weakly constrained) and thin tubes placed between columns. Please see the supplemental video for qualitative results.

## 5   Discussion

We introduce SRNs, a 3D-structured neural scene representation that implicitly represents a scene as a continuous, differentiable function. This function maps 3D coordinates to a feature-based representation of the scene and can be trained end-to-end with a differentiable ray marcher to render the feature-based representation into a set of 2D images. SRNs do not require shape supervision and can be trained only with a set of posed 2D images. We demonstrate results for novel view synthesis, shape and appearance interpolation, and few-shot reconstruction.

There are several exciting avenues for future work. SRNs could be explored in a probabilistic framework [2, 3], enabling sampling of feasible scenes given a set of observations. SRNs could be extended to model view- and lighting-dependent effects, translucency, and participating media. They could also be extended to other image formation models, such as computed tomography or magnetic resonance imaging. Currently, SRNs require camera intrinsic and extrinsic parameters, which can be obtained robustly via bundle-adjustment. However, as SRNs are differentiable with respect to camera parameters; future work may alternatively integrate them with learned algorithms for camera pose estimation [72]. SRNs also have exciting applications outside of vision and graphics, and future work may explore SRNs in robotic manipulation or as the world model of an independent agent. While SRNs can represent room-scale scenes (see the supplemental video), generalization across complex, cluttered 3D environments is an open problem. Recent work in meta-learning could enable generalization across scenes with weaker assumptions on the dimensionality of the underlying manifold [73]. Please see the supplemental material for further details on directions for future work.

## 6   Acknowledgements

We thank Ludwig Schubert for fruitful discussions. Vincent Sitzmann was supported by a Stanford Graduate Fellowship. Michael Zollhöfer was supported by the Max Planck Center for Visual Computing and Communication (MPC-VCC). Gordon Wetzstein was supported by NSF awards (IIS 1553333, CMMI 1839974), by a Sloan Fellowship, by an Okawa Research Grant, and a PECASE.

## Footnotes

[1]Please see supplemental video for additional results.

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
