[Supplementary Material]

# Scene Representation Networks: Continuous 3D-Structure-Aware Neural Scene Representations –Supplementary Material–

**Vincent Sitzmann**     **Michael Zollhöfer**     **Gordon Wetzstein**
{sitzmann, zollhoefer}@cs.stanford.edu, gordon.wetzstein@stanford.edu
Stanford University

## Contents

Raycast progress - from top left to bottom right

Final Normal Map

Raycast progress - from top left to bottom right

Final Normal Map

Figure 1: Visualizations of ray marching progress and the final normal map. Note that the uniformly colored background does not constrain the depth - as a result, the depth is unconstrained around the silhouette of the object. Since the final normal map visualizes surface detail much better, we only report the final normal map in the main document.

# 1    Additional Results on Neural Ray Marching

**Computation of Normal Maps**    We found that normal maps visualize fine surface detail significantly better than depth maps (see Fig. 1), and thus only report normal maps in the main submission. We compute surface normals as the cross product of the numerical horizontal and vertical derivatives of the depth map.

**Ray Marching Progress Visualization**    The $z$-coordinates of running and final estimates of intersections in each iteration of the ray marcher in camera coordinates yield depth maps, which visualize every step of the ray marcher. Fig. 1 shows two example ray marches, along with their final normal maps.

# 2    Comparison to DeepVoxels

We compare performance in single-scene novel-view synthesis with the recently proposed DeepVoxels architecture [1] on their four synthetic objects. DeepVoxels proposes a 3D-structured neural scene representation in the form of a voxel grid of features. Multi-view and projective geometry are hard-coded into the model architecture. We further report accuracy of the same baselines as in [1]: a Pix2Pix architecture [2] that receives as input the per-pixel view direction, as well as the methods proposed by Tatarchenko et al. [3] as well as by Worrall et al. [4] and Cohen and Welling [5].

Table 1 compares PSNR and SSIM of the proposed architecture and the baselines, averaged over all 4 scenes. We outperform the best baseline, DeepVoxels [1], by more than 3 dB. Qualitatively, DeepVoxels displays significant multi-view inconsistencies in the form of flickering artifacts, while the proposed method is almost perfectly multi-view consistent. We achieve this result with 550k parameters per model, as opposed to the DeepVoxels architecture with more than 160M free variables. However, we found that SRNs produce blurry output for some of the very high-frequency textural

Figure 2: Qualitative results on DeepVoxels objects. For each object: Left: Normal map of reconstructed geometry. Center: SRNs output. Right: Ground Truth.

Figure 3: Undersampled letters on the side of the cube (ground truth images). Lines of letters are less than two pixels wide, leading to significant aliasing. Additionally, the 2D downsampling as described in [1] introduced blur that is not multi-view consistent.

Figure 4: By using a U-Net renderer similar to [1], we can reconstruct the undersampled letters. In exchange, we lose the guarantee of multi-view consistency. Left: Reconstructed normal map. Center: SRNs output. Right: ground truth.

|                          | PSNR  | SSIM |
|--------------------------|-------|------|
| Tatarchenko et al. [3]   | 21.22 | 0.90 |
| Worrall et al. [4]       | 21.22 | 0.90 |
| Pix2Pix [2]              | 23.63 | 0.92 |
| DeepVoxels [1]           | 30.55 | 0.97 |
| SRNs                     | **33.03** | **0.97** |

Table 1: Quantitative comparison to DeepVoxels [1]. With 3 orders of magnitude fewer parameters, we achieve a 3dB boost, with reduced multi-view inconsistencies.

Figure 5: Architecture overview: at the heart of SRNs lies a continuous, 3D-aware neural scene representation, $\Phi$, which represents a scene as a function that maps $(x, y, z)$ world coordinates to a feature representation of the scene at those coordinates. To render $\Phi$, a neural ray-marcher interacts with $\Phi$ via world coordinates along camera rays, parameterized via their distance $d$ to the camera projective center. Ray Marching begins at a distance $d_0$ close to the camera. In each step, the scene representation network $\Phi$ is queried at the current world coordinates $x_i$. The resulting feature vector $v_i$ is fed to the Ray Marching LSTM that predicts a step length $\delta_{i+1}$. The world coordinates are updated according to the new distance to the camera, $d_{i+1} = d_i + \delta_{i+1}$. This is repeated for a fixed number of iterations, $n$. The features at the final world coordinates $v_n = \Phi(x_n)$ are then translated to an RGB color by the pixel generator.

detail - this is most notable with the letters on the sides of the cube. Fig. 3 demonstrates why this is the case. Several of the high-frequency textural detail of the DeepVoxels objects are heavily undersampled. For instance, lines of letters on the sides of the cube often only occupy a single pixel. As a result, the letters alias across viewing angles. This violates one of our key assumptions, namely that the same $(x, y, z) \in \mathbb{R}^3$ world coordinate always maps to the same color, independent of the viewing angle. As a result, it is impossible for our model to generate these details. We note that detail that is not undersampled, such as the CVPR logo on the top of the cube, is reproduced with perfect accuracy. However, we can easily accommodate for this undersampling by using a 2D CNN renderer. This amounts to a trade-off of our guarantee of multi-view consistency discussed in Sec. 3 of the main paper with robustness to faulty training data. Fig. 2 shows the cube rendered with a U-Net based renderer – all detail is replicated truthfully.

## 3 Reproducibility

In this section, we discuss steps we take to allow the community to reproduce our results. *All code and datasets will be made publicly available.* All models were evaluated on the test sets exactly once.

### 3.1 Architecture Details

**Scene representation network** $\Phi$    In all experiments, $\Phi$ is parameterized as a multi-layer perceptron (MLP) with ReLU activations, layer normalization before each nonlinearity [6], and four layers with 256 units each. In all generalization experiments in the main paper, its weights $\phi$ are the output of the

hypernetwork $\Psi$. In the DeepVoxels comparison (see Sec.2), where a separate $\Phi$ is trained per scene, parameters of $\phi$ are directly initialized using the Kaiming Normal method [7].

**Hypernetwork $\Psi$**    In generalization experiments, a hypernetwork $\Psi$ maps a latent vector $\mathbf{z}_j$ to the weights of the respective scene representation $\phi_j$. Each layer of $\Phi$ is the output of a separate hypernetwork. Each hypernetwork is parameterized as a multi-layer perceptron with ReLU activations, layer normalization before each nonlinearity [6], and three layers (where the last layer has as many units as the respective layer of $\Phi$ has weights). In the Shapenet and Shepard-Metzler experiments, where the latent codes $\mathbf{z}_j$ have length 256, hypernetworks have 256 units per layer. In the Basel face experiment, where the latent codes $\mathbf{z}_j$ have length 224, hypernetworks have 224 units per layer. Weights are initialized by the Kaiming Normal method, scaled by a factor $0.1$. We empirically found this initialization to stabilize early training.

**Ray marching LSTM**    In all experiments, the ray marching LSTM is implemented as a vanilla LSTM with a hidden state size of 16. The initial state is set to zero.

**Pixel Generator**    In all experiments, the pixel generator is parameterized as a multi-layer perceptron with ReLU activations, layer normalization before each nonlinearity [6], and five layers with 256 units each. Weights are initialized with the Kaiming Normal method [7].

## 3.2    Time & Memory Complexity

**Scene representation network $\Phi$**    $\Phi$ scales as a standard MLP. Memory and runtime scale linearly in the number of queries, therefore quadratic in image resolution. Memory and runtime further scale linearly with the number of layers and quadratically with the number of units in each layer.

**Hypernetwork $\Psi$**    $\Psi$ scales as a standard MLP. Notably, the last layer of $\Psi$ predicts all parameters of the scene representation $\Phi$. As a result, the number of weights scales linearly in the number of weights of $\Phi$, which is significant. For instance, with 256 units per layer and 4 layers, $\Phi$ has approximately $2 \times 10^5$ parameters. In our experiments, $\Psi$ is parameterized with 256 units in all hidden layers. The last layer of $\Psi$ then has approximately $5 \times 10^7$ parameters, which is the bulk of learnable parameters in our model. Please note that $\Psi$ only has to be queried once to obtain $\Phi$, at which point it could be discarded, as both the pixel generation and the ray marching only need access to the predicted $\Phi$.

**Differentiable Ray Marching**    Memory and runtime of the differentiable ray marcher scale linearly in the number of ray marching steps and quadratically in image resolution. As it queries $\Phi$ repeatedly, it also scales linearly in the same parameters as $\Phi$.

**Pixel Generator**    The pixel generator scales as a standard MLP. Memory and runtime scale linearly in the number of queries, therefore quadratic in image resolution. Memory and runtime further scale linearly with the number of layers and quadratically with the number of units in each layer.

## 3.3    Dataset Details

**Shepard-Metzler objects**    We modified an open-source implementation of a Shepard-Metzler renderer (https://github.com/musyoku/gqn-dataset-renderer.git) to generate meshes of Shepard-Metzler objects, which we rendered using Blender to have full control over camera intrinsic and extrinsic parameters consistent with other presented datasets.

**Shapenet v2 cars**    We render each object from random camera perspectives distributed on a sphere with radius 1.3 using Blender. We disabled specularities, shadows and transparencies and used environment lighting with energy 1.0. We noticed that a few cars in the dataset were not scaled optimally, and scaled their bounding box to unit length. A few meshes had faulty vertices, resulting in a faulty bounding box and subsequent scaling to a very small size. We discarded those 40 out of 2473 cars.

**Shapenet v2 chairs**  We render each object from random camera perspectives distributed on a sphere with radius 2.0 using Blender. We disabled specularities, shadows and transparencies and used environment lighting with energy 1.0.

**Faces dataset**  We use the Basel Face dataset to generate meshes with different identities at random, where each parameter is sampled from a normal distribution with mean 0 and standard deviation of 0.7. For expressions, we use the blendshape model of Thies et al. [8], and sample expression parameters uniformly in $(-0.4, 1.6)$.

**DeepVoxels dataset**  We use the dataset as presented in [1].

### 3.4  SRNs Training Details

#### 3.4.1  General details

**Multi-Scale training**  Our per-pixel formulation naturally allows us to train in a coarse-to-fine setting, where we first train the model on downsampled images in a first stage, and then increase the resolution of images in stages. This allows larger batch sizes at the beginning of the training, which affords more independent views for each object, and is reminiscent of other coarse-to-fine approaches [9].

**Solver**  For all experiments, we use the ADAM solver with $\beta_1 = 0.9$, $\beta_2 = 0.999$.

**Implementation & Compute**  We implement all models in PyTorch. All models were trained on single GPUs of the type RTX6000 or RTX8000.

**Hyperparameter search**  Training hyperparameters for SRNs were found by informal search – we did not perform a systematic grid search due to the high computational cost.

#### 3.4.2  Per-experiment details

For a resolution of $64 \times 64$, we train with a batch size of 72. Due to the memory complexity being quadratic in the image sidelength, we decrease the batch size by a factor of 4 when we double the image resolution. $\lambda_\text{depth}$ is always set to $1 \times 10^{-3}$ and $\lambda_\text{latent}$ is set to 1. The ADAM learning rate is set to $4 \times 10^{-4}$ if not reported otherwise.

**Shepard-Metzler experiment**  We directly train our model on images of resolution $64 \times 64$ for 352 epochs.

**Shapenet cars**  We train our model in 2 stages. We first train on a resolution of $64 \times 64$ for 5k iterations. We then increase the resolution to $128 \times 128$. We train on the high resolution for 70 epochs. The ADAM learning rate is set to $5 \times 10^{-5}$.

**Shapenet chairs**  We train our model in 2 stages. We first train on a resolution of $64 \times 64$ for 20k iterations. We then increase the resolution to $128 \times 128$. We train our model for 12 epochs.

**Basel face experiments**  We train our model in 2 stages. We first train on a resolution of $64 \times 64$ for 15k iterations. We then increase the resolution to $128 \times 128$ and train for another 5k iterations.

**DeepVoxels experiments**  We train our model in 3 stages. We first train on a resolution of $12 \times 128$ with a learning rate of $4 \times 10^{-4}$ for 20k iterations. We then increase the resolution to $256 \times 256$, and lower the learning rate to $1 \times 10^{-4}$ and train for another 30k iterations. We then increase the resolution to $512 \times 512$, and lower the learning rate to $4 \times 10^{-6}$ and train for another 30k iterations.

## 4  Relationship to per-pixel autoregressive methods

With the proposed per-pixel generator, SRNs are also reminiscent of autoregressive per-pixel architectures, such as PixelCNN and PixelRNN [10, 11]. The key difference to autoregressive per-pixel

architectures lies in the modeling of the probability $p(\mathcal{I})$ of an image $\mathcal{I} \in \mathbb{R}^{H \times W \times 3}$. PixelCNN and PixelRNN model an image as a one-dimensional sequence of pixel values $\mathcal{I}_1, ..., \mathcal{I}_{H \times W}$, and estimate their joint distribution as

$$p(\mathcal{I}) = \prod_{i=1}^{H \times W} p(\mathcal{I}_i | \mathcal{I}_1, ..., \mathcal{I}_{i-1}). \tag{1}$$

Instead, conditioned on a scene representation $\Phi$, pixel values are conditionally independent, as our approach independentaly and deterministically assigns a value to each pixel. The probability of observing an image $\mathcal{I}$ thus simplifies to the probability of observing a scene $\Phi$ under extrinsic $\mathbf{E}$ and intrinsic $\mathbf{K}$ camera parameters

$$p(\mathcal{I}) = p(\Phi)p(\mathbf{E})p(\mathbf{K}). \tag{2}$$

This conditional independence of single pixels conditioned on the scene representation further motivates the per-pixel design of the rendering function $\Theta$.

## 5 Baseline Discussions

### 5.1 Deterministic Variant of GQN

**Deterministic vs. Non-Deterministic**   Eslami et al. [12] propose a powerful probabilistic framework for modeling uncertainty in the reconstruction due to incomplete observations. However, here, we are exclusively interested in investigating the properties of the scene representation itself, and this submission discusses SRNs in a purely deterministic framework. To enable a fair comparison, we thus implement a deterministic baseline inspired by the Generative Query Network [12]. We note that the results obtained in this comparison are not necessarily representative of the performance of the unaltered Generative Query Network. We leave a formulation of SRNs in a probabilistic framework and a comparison to the unaltered GQN to future work.

**Architecture**   As representation network architecture, we choose the "Tower" representation, and leave its architecture unaltered. However, instead of feeding the resulting scene representation $\mathbf{r}$ to a convolutional LSTM architecture to parameterize a density over latent variables $\mathbf{z}$, we instead directly feed the scene representation $\mathbf{r}$ to a generator network. We use as generator a deterministic, autoregressive, skip-convolutional LSTM $C$, the deterministic equivalent of the generator architecture proposed in [12]. Specifically, the generator can be described by the following equations:

| | | |
|---|---|---|
| Initial state | $(\mathbf{c}_0, \mathbf{h}_0, \mathbf{u}_0) = (\mathbf{0}, \mathbf{0}, \mathbf{0})$ | (3) |
| Pre-process current canvas | $\mathbf{p}_l = \kappa(\mathbf{u}_l)$ | (4) |
| State update | $(\mathbf{c}_{l+1}, \mathbf{h}_{l+1}) = C(\mathbf{E}, \mathbf{r}, \mathbf{c}_l, \mathbf{h}_l, \mathbf{p}_l)$ | (5) |
| Canvas update | $\mathbf{u}_{l+1} = \mathbf{u}_l + \Delta(\mathbf{h}_{l+1})$ | (6) |
| Final output | $\mathbf{x} = \eta(\mathbf{u}_L),$ | (7) |

with timestep $l$ and final timestep $L$, LSTM output $\mathbf{c}_l$ and cell $\mathbf{h}_l$ states, the canvas $\mathbf{u}_l$, a downsampling network $\kappa$, the camera extrinsic parameters $\mathbf{E}$, an upsampling network $\Delta$, and a $1 \times 1$ convolutional layer $\eta$. Consistent with [12], all up- and downsampling layers are convolutions of size $4 \times 4$ with stride 4. To account for the higher resolution of the Shapenet v2 car and chair images, we added a further convolutional layer / transposed convolution where necessary.

**Training**   On both the cars and chairs datasets, we trained for $180,000$ iterations with a batch size of $140$, taking approximately $6.5$ days. For the lower-resolution Shepard-Metzler objects, we trained for $160,000$ iterations at a batch size of $192$, or approximately $5$ days.

**Testing**   For novel view synthesis on the training set, the model receives as input the 15 nearest neighbors of the novel view in terms of cosine similarity. For two-shot reconstruction, the model receives as input whichever of the two reference views is closer to the novel view in terms of cosine similarity. For one-shot reconstruction, the model receives as input the single reference view.

Encoder

Decoder

Image

From feature
transform

| 128x128x3 | conv3x3/1 |
| 128x128x64 | conv4x4/2 |
| 64x64x128 | conv3x3/1 |
| 64x64x128 | conv4x4/2 |
| 32x32x256 | conv3x3/1 |
| 32x32x256 | conv4x4/2 |
| 16x16x256 | conv3x3/1 |
| 16x16x256 | conv4x4/2 |
| 8x8x256 | conv3x3/1 |
| 8x8x256 | conv4x4/2 |
| 4x4x256 | conv3x3/1 |
| 4x4x256 | conv4x4/2 |
| 4x4x256 | fc 2048 |
| 2048 | fc 2048 |
| 2048 | fc 1900*3 |

fc 1900*3       To feature
transform

| 3x1900 | fc 2048 |
| 2048 | fc 2048 |
| 2048 | fc 4x4x256 |
| 2x NN upsampling | |
| 8x8x256 | conv3x3/1 |
| 8x8x256 | conv3x3/1 |
| 2x NN upsampling | |
| 16x16x256 | conv3x3/1 |
| 16x16x256 | conv3x3/1 |
| 2x NN upsampling | |
| 32x32x256 | conv3x3/1 |
| 32x32x256 | conv3x3/1 |
| 2x NN upsampling | |
| 64x64x128 | conv3x3/1 |
| 64x64x128 | conv3x3/1 |
| 2x NN upsampling | |
| 128x128x64 | conv3x3/1 |
| 128x128x64 | conv3x3/1 |
| 128x128x3 | conv3x3/1 |
| | TanH |

Novel View

| Fully Connected + LeakyReLU |
| • + BatchNorm + LeakyReLU |

Figure 6: Architecture of the baseline method proposed in Worrall et al. [4].

## 5.2 Tatarchenko et al.

**Architecture**   We implement the exact same architecture as described in [3], with approximately $70 \cdot 10^6$ parameters.

**Training**   For training, we choose the same hyperparameters as proposed in Tatarchenko et al. [3]. As we assume no knowledge of scene geometry, we do not supervise the model with a depth map. As we observed the model to overfit, we stopped training early based on model performance on the held-out, official Shapenet v2 validation set.

**Testing**   For novel view synthesis on the training set, the model receives as input the nearest neighbor of the novel view in terms of cosine similarity. For two-shot reconstruction, the model receives as input whichever of the two reference views is closer to the novel view. Finally, for one-shot reconstruction, the model receives as input the single reference view.

## 5.3 Worrall et al.

**Architecture**   Please see Fig. 6 for a visualization of the full architecture. The design choices in this architecture (nearest-neighbor upsampling, leaky ReLU activations, batch normalization) were made in accordance with Worrall et al. [4].

**Training**   For training, we choose the same hyperparameters as proposed in Worrall et al. [4].

**Testing**   For novel view synthesis on the training set, the model receives as input the nearest neighbor of the novel view in terms of cosine similarity. For two-shot reconstruction, the model receives as input whichever of the two reference views is closer to the novel view. Finally, for one-shot reconstruction, the model receives as input the single reference view.

# 6 Differentiable Ray-Marching in the context of classical renderers

The proposed neural ray-marcher is inspired by the classic sphere tracing algorithm [13]. Sphere tracing was originally developed to render scenes represented via analytical signed distance functions. It is defined by a special choice of the step length: each step has a length equal to the signed distance to the closest surface point of the scene. Since this distance is only zero on the surface of the scene, the algorithm takes non-zero steps until it has arrived at the surface, at which point no further steps are taken. A major downside of sphere-tracing is its weak convergence guarantee: Sphere tracing is only guaranteed to converge for an infinite number of steps. This is easy to see: For any fixed number of steps, we can construct a scene where a ray is parallel to a close surface (or falls through a slim tunnel) and eventually intersects a scene surface. For any constant number of steps, there exists a surface parallel to the ray that is so close that the ray will not reach the target surface. In classical sphere-tracing, this is circumvented by taking a large number of steps that generally take the intersection estimate within a small neighborhood of the scene surface – the color at this point is then simply defined as the color of the closest surface. However, this heuristic can still fail in constructed examples such as the one above. Extensions of sphere tracing propose heuristics to modifying the step length to speed up convergence [11]. The Ray-Marching LSTM instead has the ability to learn the step length. The key driver of computational and memory cost of the proposed rendering algorithm is the ray-marching itself: In every step of the ray-marcher, for every pixel, the scene representation $\phi$ is evaluated. Each evaluation of $\phi$ is a full forward pass through a multi-layer perceptron. See 3.2 for an exact analysis of memory and computational complexity of the different components.

Other classical rendering algorithms usually follow a different approach. In modern computer graphics, scenes are often represented via explicit, discretized surface primitives - such as is the case in meshes. This allows rendering via rasterization, where scene geometry is projected onto the image plane of a virtual camera in a single step. As a result, rasterization is computationally cheap, and has allowed for real-time rendering that has approached photo-realism in computer graphics.

However, the image formation model of rasterization is not appropriate to simulate physically accurate image formations that involve proper light transport, view-dependent effects, participating media, refraction, translucency etc. As a result, physics-based rendering usually uses ray-tracing algorithms, where for each pixel, a number of rays are traced from the camera via all possible paths to light sources through the scene. If the underlying scene representations are explicit, discrete representations – such as meshes – the intersection testing required is again cheap. Main drivers of computational complexity in such systems are then the number of rays that need to be traced to appropriately sample all paths to lights sources that contribute to the value of a single pixel.

In this context, the proposed ray-marcher can be thought of as a sphere-tracing-inspired ray-tracer for implicitly defined scene geometry. It does not currently model multi-bounce ray-tracing, but could potentially be extended in the future (see 8).

# 7 Trade-offs of the Pixel Generator vs. CNN-based renderers

As described in the main paper, the pixel generator comes with a guarantee of multi-view consistency compared to a 2D-CNN based rendering network. On the flip side, we cannot make use of progress in the design of novel CNN architectures that save memory by introducing resolution bottlenecks and skip connections, such as the U-Net [14]. This means that the pixel generator is comparably memory-hungry, as each layer operates on the full resolution of the image to be generated. Furthermore, CNNs have empirically been demonstrated to be able to generate high-frequency image detail easily. It is unclear what the limitations of the proposed pipeline are with respect to generating high-frequency textural detail. We note that the pixel generator is not a necessary component of SRNs, and can be replaced by a classic 2D-CNN based renderer, as we demonstrate in 2.

# 8 Future work

**Applications outside of vision.** SRNs have promising applications outside of vision. Neural scene representations are a core aspect of artificial intelligence, as they allow an agent to model its environment, navigate, and plan interactions. Thus, natural applications of SRNs lie in robotic manipulation or as the world model of an independent agent.

**Extending SRNs to other image formation models.** SRNs could be extended to other image formation models, such as computer tomography or magnetic resonance imaging. All that is required is a differentiable forward model of the image formation. The ray-marcher could be adapted accordingly to integrate features along a ray or to sample at pre-defined locations. For image formation models that observe scenes directly in 3D, the ray-marcher may be left out completely, and $\phi$ may be sampled directly.

**Probabilistic formulation.** An interesting avenue of future work is to extend SRNs to a probabilistic model that can infer a probability distribution over feasible scenes consistent with a given set of observations. In the following, we formulate one such approach, very similar to the formulation of Kumar et al. [15], which is in turn based on the work of Eslami et al. [12]. Please note that this formulation is not experimentally verified in the context of SRNs and is described here purely to facilitate further research in this direction.

Formally, the model can be summarized as:

$$r_i = M(\mathcal{I}_i, \mathbf{E}_i, \mathbf{K}_i) \tag{8}$$

$$r = \sum_i r_i \tag{9}$$

$$z \sim P_\Theta(z|r) \tag{10}$$

$$\phi = \Psi(z) \tag{11}$$

$$\mathcal{I} = \Theta(\Phi_\phi, \mathbf{E}, \mathbf{K}) \tag{12}$$

We assume that we are given a set of instance datasets $\mathcal{D} = \{\mathcal{C}_j\}_{j=1}^M$, where each $\mathcal{C}_j$ consists of tuples $\{(\mathcal{I}_i, \mathbf{E}_i, \mathbf{K}_i)\}_{i=1}^N$. For a single scene $\mathcal{C}$ with $n$ observations, we first replicate and concatenate the camera pose $\mathbf{E}_i$ and intrinsic parameters $\mathbf{K}_i$ of each observations to the image channels of the corresponding 2D image $\mathcal{I}_i$. Using a learned convolutional encoder $M$, we encode each of the $n$ observations to a code vector $r_i$. These code vectors $r_i$ are then summed to form a permutation-invariant representation of the scene $r$. Via an autoregressive DRAW model [16], we form a probability distribution $P_\theta$ that is conditioned on the code vector $r$ and sample latent variables $z$. $z$ is decoded into the parameters of a scene representation network, $\phi$, via a hypernetwork $\Psi(z) = \phi$. Lastly, via our differentiable rendering function $\Theta$, we can render images $\mathcal{I}$ from $\Phi_\phi$ as described in the main paper. This allows to train the full model end-to-end given only 2D images and their camera parameters. We note that the resulting optimization problem is intractable and requires the optimization of an evidence lower bound via an approximate posterior, which we do not derive here – please refer to [15]. Similarly to [15], this formulation will lead to multi-view consistent renderings of each scene, as the scene representation $\Phi$ stays constant across queries of $\Theta$.

**View- and lighting-dependent effects, translucency, and participating media.** Another exciting direction for future work is to model further aspects of realistic scenes. One such aspect is view- and lighting dependent effects, such as specularities. For fixed lighting, the pixel generator could receive as input the direction of the camera ray in world coordinates, and could thus reason about the view-dependent color of a surface. To model simple lighting-dependent effects, the pixel generator could further receive the light ray direction as an input (assuming no occlusions). Lastly, the proposed formulation could also be extended to model multiple ray bounces in a ray-casting framework. To model translucency and participating media, the ray-marcher could be extended to sum features along a ray instead of only sampling a feature at the final intersection estimate.

**Complex 3D scenes and compositionality.** While SRNs can represent room-scale scenes (see supplementary video), generalization across such complex, cluttered 3D environments is an open problem. To the best of our knowledge, is has not yet been demonstrated that low-dimensional embeddings are a feasible representation for photo-realistic, general 3D environments. Recent work in meta-learning could enable generalization across scenes without the limitation to a highly low-dimensional manifold [17].