[Reviews · NeurIPS 2019]

Reviewer 1



Originality I think this submission is very original as it aims at combining the physical process or rendering a scene via ray-casting with learning scene representation via deep-learning. This is actually a very natural thing to do, and I'm happy that the authors came up with this idea and made it work. Quality This submission is of high quality. The algorithm is explained clearly and the equations make sense. The authors made the effort of re-implementing three other competing techniques and compare their results against them. Clarity The authors describe all parts of the system separately and clearly, and explain how to combine them. The equations help to understand how they plug into each other and I am happy with the level of clarity overall. Significance This is a good building block that will surely open the door to more complex models combining 3D processed and deep-learning, as well are experiments at a higher scale. I think it is significant and I can see how this work could be extended in the future.

Reviewer 2



The proposed scene representation is a map phi from the 3D physical space to a feature space encoding properties such as color, distance from closest scene surface, in practice implemented with an MLP. This choice of parametrization results in a natural way of controlling the level of spatial detail the map can achieve with the chosen network capacity, without using a fixed/discrete spatial resolution (as in voxel grids). Images are generated from the scene representation phi, conditioned on a given camara (inclusive of intrinsic and extrinsic parameters) via a differentiable ray marching algorithm. Ray marching is performed using a fixed length unroll of an RNN which can operate on phi effectively decoding distance from the closest surface and learning to correctly how to update the marcher step length. This formulation has the nice bi-product of producing depth maps 'for free'. The output image is finally decoded from phi, sampled along camera rays according to the ray marcher predictions, independently per pixel using another MLP. The parameters of phi are themselves the output of a (hypernetwork) function mapping a low dimensional latent vector z to scene function. The authors propose to learn a 'per class' hypernetwork, so that z effectively becomes a per object vector. The raymarcher parameters, the hypernetwork, and z all all optimized jointly. If the hypernet and the raymarcher for a given class have been previously trained, embeddings z for a new object instance are recovered via gradient descent. If the z are known a priori (for example the parameters of CG model), by training the rest of the model conditioned on z, the system learns to correctly factor geometry and appearance. The experimental section of the paper is extensive and the authors compare to multiple models in the space of 3d understanding. Whilst I found the experiments convincing on showcasing the properties of SRN, I think they are not always particularly fair with their competitors. For example, when comparing with dGQN - the work I am the most familiar with - I believe there's one disclaimer missing, that is that dGQN performs amortised inference, hence is very data hungry and massively underperforms in the experiments in a low training data regime. On the other hand, SRN require to optimize z for each object instance that is presented to the system and this optimization will be very expensive, espcially becuase each parameter update will require backpropping through multiple steps of the ray marching LSTM, whose state will be at the same size as the final output image. since that the final rendering convultions are 1x1 to guarantee multiview consistency, as claimed by the authors (perhaps this could be indeed mitigated by not using 1x1 convolution and improving on the design of the final rendering step). To the best of my understanding, all the claims in the paper are backed up by convincing experiments, but one. At line 59 the paper reads 'formulation generalizes to modeling of uncertainty due to incomplete observations'; I believe this is an over-reaching statement, as the paper doesn't contain any analysis of the model uncertainty, and only refers to the auto-decoding work in the DeepSDF paper for details. In the reference I could not find a specific analysis of the model uncertainty either, but only experiments supporting the claim that DeepSDF can indeed handle partial observations (which is not equivalent to modelling uncertainty). There are a few natural questions that the author do not address in the submission, namely: * How does the model scale to multi-object scene? The learnt priors are class specific - can the model be adapted to be composable and scale to complex scenes? * The model will not handle specularities, reflections, transparent materials - could the renderer be adapted? * The paper only discusses the benefits of pixel independent decoding, what are the limitations? * In GQN they authors show that the scene representation can be semantically interpreted and manipulated (they show something like scene algebra), and enables view independent control in RL. Can the proposed scene representation be used for anything else than new view synthesis? I think that grounding the work in a broader context would massively increase the impact of the work. Although I think this is overall a very clear, I would like to offer some feedback with suggested changes of things that I found misleading/confusing: * line 11: 'end-to-end from only 2D observations' is misleading, since the techniques requires calibrated views; the abstract should reflect that main body the paper and read as line 42 'posed 2D images of a scene'. * line 71: 'with only 2D supervision in the image domain', same as line 11. * line 184: for the input tuples I would the same formalism as line 200, making it clear it's a set of tuple.

Reviewer 3



Originality: To the best of my knowledge, this is the first paper which proposes continues neural representation which is compatible with multi view / projective geometry toolbox and at the same time does not require dense 3D ground-truth data. Quality: The technical content appears to be correct. There are multiple aspects I like about this paper: + continuous representation, memory efficient while capable of operating at high spatial resolution + compatible with multiview/projective geometry + differentiable ray-casting is a contribution on its own + multiple applications + authors will release the source code and have paid a lot of attention to reproducibility and fair evaluation While the paper (resp supplementary) provide discussion about time and memory complexity, it would be good to add some real numbers into the paper. Similarly, it would be interesting to see how many iterations the differentiable raycasting requires, resp how slow/fast it is, and how it compares with standard non-differentiable algorithms in terms of speed (I'm not saying it has to be faster to be interesting and I do get the point that the differentiable one enables training of this model - I just think it would be interesting to put it more into context). Also, most experiments are carried out on very small scenes, typically consisting of a single object. It would be good to discuss how well the proposed could generalize to i) large-scale scenes and ii) unconstrained environments (ie not a single object, rather a room or some other more generic 3D scene); it might be interesting to add such experiments, using e.g. (significantly downsampled) ScanNet dataset. Clarity: While the paper is pretty readable, there is certainly room for improvements in the clarity of the paper. In particular, the paper should be self-contained - many implementation details are provided only in the supplementary. While I do understand it is difficult to fit the paper into 8 pages, it would be great to move these details into the main paper. Most parts of the paper read well, however, I believe that adding a figure illustrating the whole model (not just raycaster) would help the reader. Figure 1 is pretty clear, however, text/symbols are very small and reader has to zoom in - would be great to fix. First few lines of abstract (1-5) seem to be bit rushed and would be great to rephrase them. Similarly first paragraph of introduction (l. 17-21) do not quite match the quality of writing of the rest of the paper (deleting l. 17-21 would not make the introduction any worse). But in overall, I've enjoyed reading this paper! Significance: Exploring continuous neural representations is of great interest, as the discrete counterparts often do not scale well, often have significant memory and compute requirements and limited spatial representation. While the proposed approach was tested only on very small-scale and (relatively) simple scenes, it represents an important step forward.

[Author Response · NeurIPS 2019]

**Scene Representation Networks (SRNs): Continuous 3D-Structure-Aware Neural Scene Representations**

We are glad that the reviewers found SRNs to "represent an important step forward" (R3), be "very interpretable" (R2),
and that they will "open the door to more complex models combining 3D processing and deep-learning" (R1). We
thank the reviewers for their detailed, constructive feedback, which we incorporate as follows.

**Probabilistic formulation & uncertainty (R1, R2)** Our intention was to state that it could *in principle* be possible
to embed SRNs in a probabilistic framework. *We will clarify & soften the claim.* To encourage future work, we will
formalize an instantiation similar to "Consistent Jumpy Predictions for Videos and Scenes" (Kumar et al. 2018, follow-
up work to Eslami et al. 2018) in the supplement, *stating that this has not been experimentally verified.* High-level idea:
images & pose observations are encoded into a code vector $\mathbf{r}$, $\mathbf{r}$ is used to parameterize a prior distribution over latent
variables $\mathbf{z}$, and sampled latent variables $\mathbf{z}$ are decoded into a scene representation $\Phi$ by a hypernetwork.

**Complex scenes & compositionality (R1, R2, R3)** We'd like to disentangle SRNs from the notion of generalizing
SRNs. **(1)** In Sections 3.1 to 3.2.2, we formalize an SRN as a *single* function $\Phi$, representing a *single* scene. The
minecraft room at the end of the video demonstrates that this may represent challenging scenes. We are happy to add
more such examples. **(2)** For generalization, we demonstrate that the space spanned by SRNs allows learning strong
shape & appearance priors. We demonstrate this using hypernetworks, *assuming that scenes lie in a low-dimensional*
*subspace (Sec. 3.3.).* It is an open research question if this assumption holds for complex 3D scenes. We thus focus
generalized SRNs on single-object scenes. Other approaches to generalization, such as Model-Agnostic Meta-Learning
(Finn et al. 2018) may relax this assumption. Intuitively, it may be possible to copy and compose learned representations
of objects & primitives in later layers of an SRN by "re-wiring" earlier layers. This is an interesting avenue of future
work, outside the scope of this manuscript. We will clarify that generalization via hypernetworks is only valid if the
assumption in Sec. 3.3. holds, which we only demonstrate for single-object scenes, and discuss alternatives.

**dGQN solves harder task (R2)** We will clarify that the dGQN solves a more difficult problem, and that the auto-decoder
framework requires optimization to infer a scene representation.

**Required camera poses (R1, R2)** We will clarify that camera poses and intrinsic parameters are required, and will
clarify abstract, lines 11 and 71 to point out that poses are a form of geometric supervision.

**Absence of camera poses (R1)** Sparse bundle-adjustment provides fast pose & intrinsics estimation. Recent work also
formulates pose estimation in a learning framework (Ba-net, Tang et al., 2018). As SRNs are differentiable w.r.t. to
camera poses, they may be integrated with any such algorithm. We will add a discussion and references to Sec. 5.

**Metrics (R1, R3)** We will add forward pass duration ($\approx 120$ms), training memory requirements ($\approx 3$GB per batch
item), and training time ($\approx 6$ days for chairs, cars) in Sec. 4 (numbers for resolution $128 \times 128$, 10 raymarching steps).

**View-dependent effects, transparency (R2)** SRNs may be extended as follows: specular highlights can be addressed
by supplying view direction to the renderer, transparencies by accumulating features along each ray, reflections by
introducing secondary rays. We will discuss this in Sec. 5 and add an extended discussion to the supplement.

**Limitations of pixel-independent decoding (R2)** There are two key limitations: (1) 2D CNNs perform well in
generating high-frequency patterns. The current pixel generator cannot exploit this strength, but guarantees view
consistency. (2) A per-point formulation requires the LSTM, renderer and SRN to propagate features proportional to
the number of pixels, which is expensive. We will add this discussion of limitations to Sec. 5.

**Applications outside of vision, broader context (R2)** We think that SRNs have high potential for applications outside
of vision, such as robotics, physics modeling, and even medical imaging. In this manuscript, we chose to investigate a
single application in order to fully explore their fundamental properties. We will highlight this as part of future work
in this emerging area. We will rephrase discussion, abstract and introduction to contextualize this work with more
applications outside of vision.

**Raymarching vs. other renderers (R3)** We will add a brief discussion of other rendering techniques to Sec. 3.2.1,
alluding to an in-depth discussion in the supplement.

**Full-model figure (R3)** We will rework Fig. 1 to make it more legible, more clearly label the three key aspects of
the model, and make their interaction more comprehensive. We will add an additional figure to the supplement that
illustrates SRNs without spatial constraints.

**Re-work abstract, introduction (R3)** We will rephrase lines 1-5 and lines 17-21 to provide a more contextualized
introduction to the problem of scene representations.

**Expanding discussion / implementation details (R1, R3)** We agree that many details from the supplement would
significantly add to the main paper. *We will follow all such requests made in the summary of review.* However, the
manuscript is at the page limit. Adding anything will require moving something else into the supplement. We believe
that all current content is important to communicate key aspects of SRNs.

[Meta-Review · NeurIPS 2019]

All reviewers agree that the paper is of high quality and will be of interest to the NeurIPS community. I look forward to reading the camera ready version of the manuscript that takes into account the reviewers' suggestions on writing clarity and exposition.